# Urinary Sodium and Potassium Levels and Blood Pressure in Population with High Sodium Intake

**DOI:** 10.3390/nu12113442

**Published:** 2020-11-10

**Authors:** Da Young Song, Jiyoung Youn, Kyunga Kim, Joohon Sung, Jung Eun Lee

**Affiliations:** 1Department of Food and Nutrition, Sookmyung Women’s University, Seoul 04310, Korea; hanasdy@gmail.com; 2Department of Food and Nutrition, Seoul National University, Seoul 08826, Korea; jiyoungyoun46@gmail.com; 3Statistics and Data Center, Research Institute for Future Medicine, Samsung Medical Center, Seoul 06351, Korea; kyunga.j.kim@samsung.com; 4Department of Digital Health, The Samsung Advanced Institute for Health Sciences & Technology (SAIHST), Sungkyunkwan University, Seoul 06335, Korea; 5Department of Epidemiology, Graduate School of Public Health, Seoul National University, Seoul 08826, Korea; jsung@snu.ac.kr; 6Institute of Health & Environment, Seoul National University, Seoul 08826, Korea; 7Research Institute of Human Ecology, Seoul National University, Seoul 08826, Korea

**Keywords:** sodium, potassium, systolic blood pressure, diastolic blood pressure, Healthy Twin Study

## Abstract

The purpose of this study was to examine the association of urinary sodium-to-creatinine ratio and potassium-to-creatinine ratio with blood pressure in a cross-sectional study comprising Korean adults who participated in the Healthy Twin Study. The participants consisted of 2653 men and women in the Healthy Twin Study aged ≥19 years. Participants’ urinary excretion of sodium, potassium, and creatinine was measured from overnight half-day urine samples. Food intake was assessed using a validated food frequency questionnaire. We examined systolic and diastolic blood pressures according to sodium- or potassium-to-creatinine ratios using the generalized linear model. We determined food groups explaining high urinary sodium- or potassium-to-creatinine ratio using the reduced rank regression and calculated sodium- or potassium-contributing food score. We observed that systolic blood pressure was higher among men and women in the highest quintile of urinary sodium-to-creatinine ratio or sodium-to-potassium ratio than it was in the lowest quintile. Geometric means (95% CIs) of the lowest and the highest quintiles of systolic blood pressure (mmHg) were 113.4 (111.8–115.0) and 115.6 (114.1–117.2; *P* for trend = 0.02), respectively, for sodium-to-creatinine ratio. The association between urinary sodium-to-creatinine and systolic blood pressure was more pronounced among individuals whose body mass index (BMI) was less than 25 kg/m^2^ (*P* for interaction = 0.03). We found that vegetables, kimchi and seaweed intake contributed to high sodium intake and a sodium-contributing food score were associated with increased blood pressure. In our study, we identified the food groups contributing to high sodium intake and found that high urinary sodium levels were associated with increasing blood pressure among Korean adults.

## 1. Introduction

Hypertension is a major risk factor for cardiovascular disease [1], some cancer types [2,3,4], and diabetes [5]. Global Burden of Disease reported that diet may contribute to cardiovascular disease death by 52.9% [6]. Dietary factors are strongly associated with hypertension [7,8], and a reduction in sodium intake has been regarded as a promising strategy to prevent hypertension [8,9]. The Korea National Health and Nutrition Examination Survey (KNHANES) in 2018 reported 28.3% of the Korean adults aged over 30 years had hypertension [10]. The prevalence of hypertension was similar to that in American adults aged over 18 years (29.0% in the NHANES 2015–2016) [11]. Given the high sodium intake in Korea, with the mean intake being 3255.0 mg/d in 2018, which is more than 1.5 times the recommended amount of World Health Organization [10], reduction in sodium intake has been a major public health message to decrease the risk of chronic disease in Korea. 

Dietary assessment of sodium intake tends to underestimate the actual sodium intake [12] because sodium is added in cooking or at the table while eating and discretionary sodium intake is higher than nondiscretionary sodium intake in Koreans [13]. Therefore, measuring 24-h urinary sodium excretion was regarded as a desirable method for estimating sodium intake [14] and for disease association studies. As it is challenging to collect 24-h urine samples and extract urinary sodium in a large population, spot urine or half-day urine specimen are often collected.

The associations between urinary sodium levels and hypertension or blood pressure have been studied in several Western [15,16,17,18] and Korean cross-sectional studies [19,20,21,22,23]. These Korean studies including 148 to 19,476 participants found an increase in both systolic and diastolic blood pressure in individuals with high sodium excretion. A few randomized controlled trials including Dietary Approaches to Stop Hypertension (DASH) [9] and Trials of Hypertension Prevention Collaborative Research Group (TOHP) [24] found that sodium intake reduction lowered blood pressure levels. Also, several randomized controlled trials suggest the evidence that potassium supplementation decreased blood pressure [25].

In this study of more than 2000 adult participants, we examined the association between the sodium-to-creatinine ratio and potassium-to-creatinine ratio in urine and blood pressure in a cross-sectional study of Korean adults participating in the Healthy Twin Study, a large cohort study of twin families. We also determined food groups contributing to the higher sodium-to-creatinine ratio in urine, calculated the reduced rank regression (RRR) pattern score, and examined their associations with systolic and diastolic blood pressures.

## 2. Materials and Methods 

### 2.1. Study Population

The Healthy Twin Study comprised of same-sex adult twin pairs aged 30 years or older and their first-degree family members; these participants were the ones who volunteered in response to a nationwide media advertisement and mailing campaign open to the general population. From August 2005 to January 2012, a total of 3320 study participants (1333 men and 1987 women) have been enrolled. Study design and methodology have been described elsewhere [26]. 

When we examined the association of blood pressure with urinary concentrations of sodium and potassium; we excluded participants if they were not adults (under age of 19 years; *n* = 9), did not have information on blood pressure (*n* = 8), or did not have measurements of urinary sodium/potassium or creatinine (*n* = 275). We also excluded those who had been diagnosed with hypertension (*n* = 324) or cancer (*n* = 51) to avoid the potential effect of low sodium intake among participants with previous illnesses. Consequently, a total of 1034 men and 1619 women were included in the analysis of urinary sodium and potassium. 

In the analysis of food groups and blood pressure, we excluded study participants who did not answer questions pertaining to rice items or left questions for more than 50 items unanswered on the food frequency questionnaire (FFQ) (*n* = 164), as well as those whose total energy intake was beyond three standard deviations from the log_e_-transformed mean energy intake (*n* = 27). Consequently, a total of 961 men and 1501 women were included in the analysis. The study was approved by the ethics committees at the Samsung Medical Center (IRB No. 2005-08-113), Busan Paik Hospital (IRB No. 05-037), and Dankook University Hospital (IRB No. 2005-6), and written informed consent was obtained from all study participants. 

### 2.2. Ascertainment of Blood Pressure and Urinary Sodium, Potassium and Creatinine Excretion

Blood pressure measurements were performed twice with a standard manual sphygmomanometer with participants in a sitting position after 5 min rest and the average of two readings was used. 

Participants voided all urine at 7 pm and collected urine in plastic containers until the next day morning. They recorded the start and the end time of urine collection. All participants collected urine for more than eight hours. We also considered creatinine concentrations to adjust for different volume. When we adjusted for hours of urine collection, the results did not differ. Urinary sodium and potassium after >8 h of urine collection were measured using indirect ion-selective electrodes, and urinary creatinine concentration was measured using Jaffè reaction on Siemens Advia 1800 analyzers (ADVIA1800, Siemens, Malvern, PA, USA). We collected overnight urine because participant burden is lower compared to 24-h urine collection and sodium from overnight urine has been considered as an alternative approach to replace 24-h urine [27]. In our previous validation study, we reported high correlation coefficient (*r* = 0.837) between sodium excretion directly from 24-h urine and estimated sodium excretion from overnight half-day urine using Kawasaki formula in 44 participants [28]. We did not use Kawasaki formulas in the main analysis because the estimated 24-h sodium excretion may be overestimated [29]. However, on performing a sensitivity analysis using Kawasaki formulas, the results were similar. 

### 2.3. Assessment of Dietary and Non-Dietary Factors

Food intake of each participant was assessed using a validated semi-quantitative food-frequency questionnaire [30]. Participants were asked how frequently they consumed each food item during the previous year, and the frequencies were classified into the following nine categories: never, 1 time per month, 2–3 times per month, 1–2 times per week, 3–4 times per week, 5–6 times per week, 1 time per day, 2 times per day, and 3 times per day. Responses on frequencies of a specified serving size for each food item were converted to average daily intake. We consolidated 106 food items into 32 food groups based on their macronutrient composition and food preparation. Foods that did not match with a group or individual foods (e.g., nut, eggs, and coffee) remained as individual categories. 

Participants provided information on smoking status, alcohol consumption, education level, marital status, and medical history and menopausal status in women through a questionnaire. Additionally, a trained and experienced interviewer conducted face-to-face interview to clarify incomplete or equivocal responses. Each participant visited one of the centers to undergo physical examination, biochemical assessment, and anthropometric measurements. We calculated the body mass index (BMI) by dividing the weight (kg) by the height squared (m^2^). Pack-years smoked were calculated by multiplying the number of years smoked by the average number of cigarettes smoked per day. A woman was considered as postmenopausal if she did not experience menstruation for more than 12 months. If menopausal status was not reported (*n* = 33), we defined a woman as postmenopausal if the age was ≥50 years (average menopausal age in Korean women) when menopausal status was included as a covariate. The estimated glomerular filtration rate (eGFR) was determined based on the equation developed by Chronic Kidney Disease Epidemiology Collaboration (CKD-EPI) research group [31]. Circulating creatinine levels for eGFR calculation were measured using Jaffè reaction on the Siemens Advia 1800 analyzers.

### 2.4. Statistical Analysis

We calculated geometric means and 95% confidence intervals (CIs) of systolic and diastolic blood pressures according to quintiles of urinary sodium-to-creatinine ratio, potassium-to-creatinine ratio, and sodium-to-potassium ratio using the generalized linear model (GLM). We categorized the exposure by quintile for men and woman, separately. We added a random effect to the model to take family into account. We log_e_-transformed systolic and diastolic blood pressures to improve normality and exponentiated them. To test for trend across quintiles, participants were assigned the median value of their quintile level. This variable was entered as a continuous term, the coefficient for which was evaluated by the Wald test. Models were adjusted for age (years, continuous), age square (years^2^, continuous), BMI (kg/m^2^, continuous), total alcohol consumption (men: none, <15, 15–30, ≥30 g/day; women: none, <3.75, 3.75–7.5, ≥7.5 g/day), marital status (unmarried, married and living together, and divorced or widowed or separated), pack-years of smoking (men: never smoker, <10, 10–20, ≥20 pack-years; women: never smoker, <3, 3–6, ≥6 pack-years), education level (less than high school graduate, high school graduate, and college or above), and menopausal status (premenopause, postmenopause) for women. We additionally adjusted for total energy intake in food group analysis, urinary potassium (mmol/L, continuous) in sodium analysis, and urinary sodium (mmol/L, continuous) in potassium analysis. We examined whether the associations varied by BMI (<25, ≥25 kg/m^2^), smoking status (never smoker, ever smoker), alcohol consumption (non-drinker, current drinker), eGFR levels (<90, ≥90 mL/min/1.73 m^2^), urinary potassium (<37, ≥37 mmol/L, median), education level (high school graduate or college above, less than high school graduate), and marital status (married, non-married). The statistical significance of interaction was tested by using a Wald test of the beta coefficient of the cross-product term. We applied the RRR method to determine food groups explaining urinary sodium- or potassium-to-creatinine ratio [32]. RRR extracts factors that explain maximum response variation. The coefficient vectors are eigenvectors of a covariance matrix and the eigenvalue explains the variation in the corresponding linear function of predictors. The factors are sorted by decreasing eigenvalues and the first factor explains the most variation in response. We used the intake of 32 food groups as a predictor and logarithmically transformed urinary sodium excretion as responses. We derived linear combinations of 32 food groups that explained as much variation of sodium-to-creatinine ratio and potassium-to-creatinine ratio as possible. SAS PROC PLS command was used to derive the factor loading value for each food group. Food groups with an absolute factor loading of ≥0.20 were selected as groups contributing to urinary sodium excretion. The intake of each selected food group was standardized by deducing the mean and dividing the standard error (standardized Z score), and the food score was calculated by multiplying the standardized Z score by factor loading of each food group. Weighted Z-standardized food score for each food group was summed and considered as a sodium-contributing food score. We examined the association between a sodium-contributing food score and systolic and diastolic blood pressures using the GLM. All analyses were conducted using SAS version 9.4 (SAS Institute, Cary, NC, USA). Significance was set at a two-sided *P* value < 0.05.

## 3. Results

The mean overnight half-day urinary sodium excretion was 134.7 mmol/L in men and 122.0 mmol/L in women. The mean overnight half-day urinary potassium excretion was 42.3 mmol/L in men and 41.7 mmol/L in women. Baseline characteristics of participants according to the quintile of urinary sodium-to-creatinine ratio, potassium-to-creatinine ratio, and sodium-to-potassium ratio are presented in Table 1. Compared to the participants with low urinary sodium-to-creatinine ratio, those with a high ratio were more likely to be older, never smokers, divorced, widowed, or separated and less likely to have attended college or higher education. Compared to the individuals with low urinary potassium-to-creatinine levels, those with high urinary potassium were more likely to be older, never smoker, and less likely to drink alcohol and have attended college or higher education. The baseline characteristics by sex was presented in the Appendix A.

Table 2 shows the relationships of urinary sodium-to-creatinine ratio, potassium-to-creatinine ratio and sodium-to-potassium ratio with systolic and diastolic blood pressures. We observed that urinary sodium-to-creatinine ratio was positively associated with systolic blood pressure, while potassium-to-creatinine ratio was inversely associated with systolic blood pressure in men and women combined. Geometric means (95% CIs) of the lowest and the highest quintiles of systolic blood pressure (mmHg) were 113.4 (111.8–115.0) and 115.6 (114.1–117.2; *P* for trend = 0.02), respectively, for sodium-to-creatinine ratio, 115.1 (113.5–116.8) and 113.1 (111.5–114.7; *P* for trend = 0.04), respectively, for potassium-to-creatinine ratio, and 112.4 (110.9–113.9) and 115.6 (114.0–117.1; *P* for trend < 0.001), respectively, for sodium-to-potassium ratio. When we separated men and women, we observed increasing systolic blood pressure with increasing half-day urinary sodium-to-potassium ratio with systolic blood pressure in both men and women (*P* for trend < 0.05) (Appendix A). 

We examined whether the associations of urinary sodium-to-creatinine ratio and potassium-to-creatinine ratio with blood pressure differed by BMI (<25, ≥25 kg/m^2^), smoking status (never smoker, ever smoker), alcohol consumption (none, current) (Table 3), eGFR (<90, ≥90 mL/min/1.73 m^2^), or urinary potassium (<37, ≥37 mmol/L, median) (Appendix A). 

When we fitted RRR to determine food groups contributing to a high urinary sodium-to-creatinine ratio, we found the intake of vegetables, soy paste, kimchi (Korean-style fermented vegetables), and seaweed to be positively and the intake of poultry and pizza to be inversely associated with high urinary sodium-to-creatinine ratio (Table 4). A higher sodium-contributing food score was associated with increasing systolic and diastolic blood pressure in men and women combined (Table 5). The associations of higher sodium-contributing food score with systolic or diastolic blood pressure did not vary by BMI (<25, ≥25 kg/m^2^) (Appendix A). Table 6 shows the associations of selected food groups with blood pressures. There were significant associations of systolic and diastolic blood pressures with intakes of vegetables, kimchi, and seaweed (*P* for trend < 0.03). We found positive association for intakes of vegetables, potato and sweet potato, and fruit, but negative association for intakes of ramen, soft drink, poultry, red meat and beef soup, processed meat, noodle, and pizza, with urinary potassium-to-creatinine ratio (Appendix A). Potassium-contributing food score was not associated with blood pressure (Appendix A). 

## 4. Discussion

We observed that urinary sodium-to-creatinine and sodium-to-potassium ratios were positively associated with systolic blood pressure and potassium-to-creatinine ratio was inversely associated with systolic blood pressure in 2653 men and women aged 19 years or older. We found more pronounced positive associations for sodium-to-creatinine ratio in the participants with BMI < 25 kg/m^2^, never smokers, or current alcohol drinkers. We identified food groups contributing to urinary sodium-to-creatinine ratio. Intake of vegetables, kimchi, and seaweed were identified to explain the variability of urinary sodium-to-creatinine ratio and were positively associated with increased blood pressure. 

Collecting spot urine samples reduces the low compliance problem in large population study. Therefore, large epidemiologic studies often used 12-h urine replaced for 24-h urine and several studies suggested that 12-h urine sodium excretion may reflect dietary sodium intake [33]. For example, estimated correlation coefficient between mean 24-h and mean overnight sodium excretion was 0.72 in 142 male participants [27,34]. 

Our findings of positive association between urinary sodium-to-creatinine ratio and blood pressure are consistent with previous studies, which have reported a positive association between urinary sodium concentration and blood pressure in general populations. In a cross-sectional, community-based study, the 24-h urine sodium-to-potassium ratio was linearly associated with nighttime systolic blood pressure (beta-coefficient = 0.142 in linear regression) and diastolic blood pressure (beta-coefficient = 0.144) in 217 participants aged ≥ 55 years [23]. In the Korea National Health and Nutrition Examination Survey (KNHANES) 2009–2010, systolic blood pressure levels were 122.9 ± 17.4 mmHg in the top quartile but 114.3 ± 13.3 mmHg in the bottom quartile of urinary sodium-to-creatinine ratio assessed from spot urine analysis in men [35]. In the Korea Genome and Epidemiology Study (KoGES) cohort, KoGES Ansan and Ansung study, the calculated 24-h urinary sodium levels had positively linear association with systolic and diastolic blood pressure levels (*P* for trend < 0.001) [22]. The Norfolk cohort of the European prospective investigation into cancer (EPIC–Norfolk) study showed that high urinary sodium-to-creatinine ratio measured from casual urine samples was positively associated with the systolic and diastolic blood pressure [36]. Differences in systolic and diastolic blood pressures were 7.2 mmHg and 3.0 mmHg between the top and bottom quintiles of urinary sodium-to-creatinine ratio, respectively. The Melbourne Collaborative Cohort Study collected 24-h urine samples from 587 participants and found that an increment of urinary sodium of 100mmol/day was associated with 2.3 mmHg increase in systolic blood pressure [37]. In a meta-analysis of randomized trials, a reduction of 100 mmol/day in salt intake led to a decrease in systolic/diastolic blood pressure by 7.11/3.88 mmHg in hypertensive individuals and 3.57/1.66 mmHg in normotensive individuals [38]. The possible mechanism by which high urinary sodium increases blood pressure may be related to renin–angiotensin system and renal function. Excessive sodium retention leads to decrease in kidney capacity in maintaining osmotic pressure of the plasma, interstitial fluid volumes, acid–base balance, and electrical activity of cells, thus resulting in increased blood pressure [39]. 

Several epidemiologic and intervention studies reported an inverse association between potassium intake or urinary potassium excretion and blood pressure. The INTERSALT study observed that systolic and diastolic blood pressure levels increased by 3.4 mmHg and 1.9 mmHg, respectively, according to 50 nmol per day of 24-h urinary potassium excretion among 10,079 participants [15]. A recent meta-analysis of randomized-controlled trials, where participants in intervention arm were provided with potassium supplements for 4 to 24 weeks, found reduction in systolic blood pressure by 4.7 mmHg and diastolic blood pressure by 3.5 mmHg [25]. Activation of sodium-chloride cotransporter has been suggested as a potential mechanism through which low potassium increases blood pressure. Low potassium stimulates the activation of sodium-chloride cotransporter in the kidney, which promotes the sodium retention and increases blood pressure [40]. Also, sodium-hydrogen exchanger type 3 in kidney may be enhanced by potassium depletion, resulting in increase in sodium retention [41]. 

The reasons for the significant interaction by BMI were not clear. The US National Health and Nutrition Examination Survey (NHANES) Epidemiologic Follow-up Study reported increase in cardiovascular disease mortality only among overweight individuals [42]. However, the subgroup analysis of DASH diet did not find any difference in low sodium effect on SBP by obesity [43]. The association between sodium intake and blood pressure warrants further investigation. 

RRR, a new method of dietary pattern analysis, identifies linear combinations of dietary intake variables that explain the variance in a set of intermediated response variables [32]. We determined food groups contributing to urinary sodium-to-creatinine ratio using RRR and the associations of these food groups with systolic or diastolic blood pressures. Intakes of vegetables, soy paste, kimchi, and seaweed were associated with increased systolic and diastolic blood pressures. Kimchi appeared to be top food contributing to high sodium intake in Korean populations [44]. One of the most common ways to cook vegetables is to blanch them with soy sauce or salt. Roasted laver is commonly seasoned with salt and sesame oil in Korea, and dried kelp has high salts content. Significant association with systolic or diastolic blood pressure observed for RRR-derived sodium score may suggest a possible association between sodium intake and hypertension. 

There are some limitations in our study. We did not take into account dietary intake of sodium or correlation between urinary sodium intake and dietary sodium intake. Measurement error in estimating food intake from FFQs could attenuate to some extent the associations for food groups contributing to urinary sodium levels. Our cross-sectional study design may not infer the temporal relationship between urinary sodium and blood pressure. Further prospective studies are warranted to replicate the association we observed. The strengths of our study include a large number of participants and measurements of overnight urine for over 2500 participants.

## 5. Conclusions

Our study included a large number of participants in Korean population whose sodium intake is high and found that sodium intake was associated with increased levels of blood pressure. Although the association between sodium intake and blood pressure has been supported by several epidemiologic studies, urinary sodium, a better indicator for sodium intake than dietary intake, has not been well studied especially in Asian populations. Our study supports that even in population with high sodium intake, blood pressure increases with sodium intake. 

Our study suggests that sodium intake and food groups with high sodium levels lead to high urinary sodium concentration and are associated with increasing blood pressure among Korean healthy adults.

## Figures and Tables

**Table 1 nutrients-12-03442-t001:** Baseline characteristics according to the sodium-to-creatinine ratio, potassium-to-creatinine ratio and sodium-to potassium ratio in men and women combined ^1.^

	Quintile 1	Quintile 2	Quintile 3	Quintile 4	Quintile 5
Sodium-to-creatinine ratio					
No. of participants	530	531	531	531	530
Sodium-to-creatinine ratio (range)	1.60–10.40	10.40–14.20	14.20–18.32	18.34–24.51	24.51–102.48
Age (years)	37.89 (11.16)	40.78 (11.66)	41.56 (11.72)	44.41 (12.47)	46.57 (13.28)
Sodium (mmol/L)	101.74 (44.62)	121.55 (47.71)	130.05 (53.62)	138.31 (52.85)	143.02 (54.41)
Potassium (mmol/L)	52.01 (27.62)	45.01 (23.12)	40.38 (20.80)	38.23 (19.71)	34.21 (17.56)
BMI (kg/m^2^)	23.69 (3.26)	23.30 (3.20)	23.20 (3.16)	23.61 (3.21)	23.45 (3.21)
Smoking status ^2^					
Never	301 (56.79)	337 (63.47)	330 (62.15)	353 (66.48)	398 (75.09)
Past	79 (14.91)	67 (12.62)	77 (14.50)	79 (14.88)	44 (8.30)
Current	138 (26.04)	111 (20.90)	115 (21.66)	89 (16.76)	81 (15.28)
Alcohol consumption ^2^					
Never	97 (18.30)	125 (23.54)	136 (25.61)	161 (30.32)	169 (31.89)
Past	41 (7.74)	59 (11.11)	51 (9.60)	38 (7.16)	44 (8.30)
Current	391 (73.77)	347 (65.35)	344 (64.78)	332 (62.52)	317 (59.81)
Education level					
Less than high school graduate	56 (10.57)	81 (15.25)	92 (17.33)	132 (24.86)	167 (31.51)
High school graduate	161 (30.38)	184 (34.65)	172 (32.39)	194 (36.53)	190 (35.85)
College or above	310 (58.49)	266 (50.09)	264 (49.72)	204 (38.42)	171 (32.26)
Marital status ^2^					
Never married	170 (32.08)	120 (22.60)	120 (22.60)	80 (15.07)	63 (11.89)
Married	342 (64.53)	389 (73.26)	376 (70.81)	410 (77.21)	401 (75.66)
Divorced or widowed or separated	16 (3.02)	21 (3.95)	34 (6.40)	40 (7.53)	63 (11.89)
Potassium-to-creatinine ratio					
No. of participants	531	530	531	532	529
Potassium-to-creatinine ratio (range)	1.2–3.46	3.46–4.46	4.46–5.6	5.61–7.25	7.26–30.17
Age (years)	37.05 (10.96)	40.80 (11.75)	42.24 (12.00)	44.29 (12.33)	46.86 (12.89)
Sodium (mmol/L)	127.82 (52.89)	128.87 (53.20)	123.81 (51.10)	126.10 (51.53)	128.09 (55.24)
Potassium (mmol/L)	32.69 (16.93)	38.21 (18.21)	41.83 (22.83)	45.55 (22.81)	51.58 (27.37)
BMI (kg/m^2^)	23.87 (3.24)	23.64 (3.13)	23.29 (3.18)	23.22 (3.39)	23.22 (3.07)
Smoking status ^2^					
Never	241 (45.39)	304 (57.36)	345 (64.97)	397 (74.62)	432 (81.66)
Past	86 (16.20)	83 (15.66)	88 (16.57)	52 (9.77)	37 (6.99)
Current	194 (36.53)	129 (24.34)	86 (16.20)	69 (12.97)	56 (10.59)
Alcohol consumption^2^					
Never	77 (14.50)	104 (19.62)	145 (27.31)	162 (30.45)	200 (37.81)
Past	33 (6.21)	54 (10.19)	52 (9.79)	53 (9.96)	41 (7.75)
Current	421 (79.28)	372 (70.19)	334 (62.90)	317 (59.59)	287 (54.25)
Education level ^2^					
Less than high school graduate	59 (11.11)	80 (15.09)	98 (18.46)	126 (23.68)	165 (31.19)
High school graduate	179 (33.71)	174 (32.83)	178 (33.52)	184 (34.59)	186 (35.16)
College or above	292 (54.99)	276 (52.08)	253 (47.65)	216 (40.60)	178 (33.65)
Marital status ^2^					
Never married	170 (32.02)	137 (25.85)	104 (19.59)	81 (15.23)	61 (11.53)
Married	344 (64.78)	363 (68.49)	397 (74.76)	403 (75.75)	411 (77.69)
Divorced or widowed or separated	14 (2.64)	30 (5.66)	29 (5.46)	46 (8.65)	55 (10.40)
Sodium-to-potassium ratio					
No. of participants	531	530	529	52	531
Sodium-to-creatinine ratio (range)	0.32–5.83	0.93–8.95	1.23–10.65	0.75–11.93	1.57–13.42
Age (years)	41.22 (11.75)	42.04 (11.98)	43.50 (13.18)	42.67 (12.67)	41.79 (12.49)
Sodium (mmol/L)	100.54 (45.82)	118.62 (47.57)	129.47 (51.04)	139.29 (50.59)	146.73 (55.79)
Potassium (mmol/L)	63.59 (29.72)	47.05 (19.03)	40.27 (15.89)	34.12 (12.58)	24.82 (10.18)
BMI (kg/m^2^)	23.38 (3.36)	23.20 (3.11)	23.28 (3.05)	23.57 (3.21)	23.83 (3.28)
Smoking status ^2^					
Never	375 (70.62)	370 (69.81)	350 (66.16)	341 (64.10)	283 (53.30)
Past	60 (11.30)	53 (10.00)	69 (13.04)	83 (15.60)	81 (15.25)
Current	84 (15.82)	93 (17.55)	98 (18.53)	98 (18.42)	161 (30.32)
Alcohol consumption ^2^					
Never	150 (28.25)	133 (25.09)	143 (27.03)	144 (27.07)	118 (22.22)
Past	49 (9.23)	49 (9.25)	59 (11.15)	34 (6.39)	42 (7.91)
Current	331 (62.34)	348 (65.66)	327 (61.81)	354 (66.54)	371 (69.87)
Education level ^2^					
Less than high school graduate	88 (16.57)	101 (19.06)	112 (21.17)	120 (22.56)	107 (20.15)
High school graduate	164 (30.89)	186 (35.09)	176 (33.27)	184 (34.59)	191 (35.97)
College or above	277 (52.17)	240 (45.28)	241 (45.56)	225 (42.29)	232 (43.69)
Marital status ^2^					
Never married	119 (22.41)	114 (21.51)	96 (18.15)	108 (20.30)	116 (21.85)
Married	382 (71.94)	391 (73.77)	389 (73.53)	380 (71.43)	376 (70.81)
Divorced or widowed or separated	29 (5.46)	25 (4.72)	44 (8.32)	40 (7.52)	36 (6.78)

Abbreviation: BMI, body mass index. ^1^ Values are means (SD) or number (percentage). ^2^ A few participants had missing values.

**Table 2 nutrients-12-03442-t002:** Multivariate-adjusted geometric means (95% CIs) ^1^ of systolic and diastolic blood pressures (mmHg) according to urinary sodium to creatinine, potassium to creatinine, or sodium to potassium ratios

	Quintile 1	Quintile 2	Quintile 3	Quintile 4	Quintile 5	*P* Trend
Sodium-to-creatinine ratio ^2^						
Range	1.6–10.4	10.4–14.2	14.2–18.3	18.3–24.5	24.5–102.5	
SBP	113.4 (111.8–115.0)	113.8 (112.4–115.3)	114.0 (112.5–115.5)	113.9 (112.3–115.4)	115.6 (114.1–117.2)	0.02
DBP	72.3 (71.3–73.4)	72.4 (71.4–73.5)	72.5 (71.4–73.5)	72.5 (71.4–73.5)	73.2 (72.1–74.3)	0.19
Potassium-to-creatinine ratio ^3^						
Range	1.2–3.5	3.5–4.5	4.5–5.6	5.6–7.3	7.3–30.2	
SBP	115.1 (113.5–116.8)	114.6 (113.0–116.2)	114.4 (112.9–115.8)	114.5 (112.9–116.0)	113.1 (111.5–114.7)	0.04
DBP	73.1 (71.9–74.3)	72.9 (71.8–74.0)	72.5 (71.4–73.5)	72.9 (71.8–73.9)	71.9 (70.8–73.0)	0.08
Sodium-to-potassium ratio						
Range	0.3–2.2	2.2–2.9	2.9–3.6	3.6–4.7	4.7–13.4	
SBP	112.4 (110.9–113.9)	113.6 (112.1–115.1)	114.6 (113.1–116.1)	114.7 (113.2–116.3)	115.6 (114.0–117.1)	<0.001
DBP	71.4 (70.4–72.4)	73.1 (72.1–74.1)	72.9 (71.9–74.0)	72.6 (71.5–73.6)	73.0 (71.8–74.1)	0.06

Abbreviation: CI, confidence interval; SBP, systolic blood pressure; DBP, diastolic blood pressure; BMI, body mass index. ^1^ Adjusted for age, age^2^, BMI (kg/m^2^; continuous), pack years of smoking (never, <10, 10 to <20, ≥20 for men and never, <3, 3 to <6, ≥6 for women), alcohol consumption(none, <15, 15 to <30, ≥30 g/day for men and none, <3.75, 3.75 to <7.5, ≥7.5 g/day for women), marriage status (never, married, divorced or widowed or separated), education level (less than high school graduate, high school graduate, college or above), and menopause status (pre, post) for women. ^2^ We further adjusted for urinary potassium (mmol/L; continuous) in the model. ^3^ We further adjusted for urinary sodium (mmol/L; continuous) in the model.

**Table 3 nutrients-12-03442-t003:** Multivariate-adjusted geometric means (95% CIs) ^1^ of systolic and diastolic blood pressures (mmHg) according to urinary sodium-to-creatinine and potassium-to-creatinine ratios by BMI, smoking status, and alcohol consumption

	*n*	Quintile 1	Quintile 2	Quintile 3	Quintile 4	Quintile 5	*P* Trend	*P* Interaction
Sodium-to-creatinine ratio ^2^								
SBP								
BMI (kg/m^2^)								
<25	1886	110.8 (108.9–112.7)	111.1 (109.4–112.8)	111.7 (110.0–113.4)	112.1 (110.3–113.9)	113.3 (111.5–115.1)	0.01	0.03
≥25	767	119.8 (116.5–123.1)	120.3 (117.5–123.2)	119.5 (116.7–122.4)	118.9 (115.9–121.9)	120.4 (117.5–123.4)	0.91	
Smoking status								
Never smoker	1719	113.1 (111.3–115.0)	113.4 (111.7–115.0)	114.1 (112.4–115.9)	113.5 (111.8–115.2)	115.3 (113.5–117.0)	0.04	0.34
Ever smoker	925	114.0 (110.7–117.4)	114.2 (111.2–117.4)	113.6 (110.5–116.7)	114.2 (111.1–117.4)	114.6 (111.3–118.1)	0.72	
Alcohol consumption								
Non-drinker	920	112.1 (109.0–115.2)	113.5 (111.1–116.0)	114.4 (111.8–117.1)	114.0 (111.4–116.7)	114.0 (111.4–116.6)	0.39	0.52
Current drinker	1732	113.8 (111.8–115.7)	113.5 (111.7–115.4)	113.5 (111.6–115.4)	113.3 (111.5–115.2)	116.2 (114.2–118.1)	0.03	
DBP								
BMI (kg/m^2^)								
<25	1886	70.6 (69.3–71.9)	71.0 (69.8–72.2)	70.8 (69.6–72.1)	71.1 (69.8–72.3)	71.4 (70.2–72.6)	0.30	0.65
≥25	767	75.9 (73.9–77.9)	75.6 (73.6–77.7)	76.0 (73.9–78.1)	76.3 (74.1–78.5)	76.9 (74.8–79.1)	0.32	
Smoking status								
Never smoker	1719	72.4 (71.1–73.7)	72.3 (71.0–73.5)	72.5 (71.2–73.8)	72.2 (70.9–73.5)	7.30 (71.7–74.3)	0.38	0.75
Ever smoker	925	72.0 (70.0–74.2)	72.3 (70.2–74.4)	71.8 (69.7–73.9)	72.3 (70.2–74.5)	72.2 (69.8–74.7)	0.87	
Alcohol consumption								
Non-drinker	920	72.4 (70.3–74.5)	72.8 (71.1–74.5)	73.7 (71.8–75.6)	73.2 (71.3–75.1)	72.5 (70.5–74.5)	0.85	0.83
Current drinker	1732	72.1 (70.9–73.4)	71.8 (70.6–73.1)	71.6 (70.3–73.0)	71.7 (70.4–73.1)	73.3 (72.0–74.6)	0.10	
Potassium-to-creatinine ratio ^3^								
SBP								
BMI (kg/m^2^)								
<25	1886	112.9 (110.9–114.9)	112.1 (110.2–113.9)	112.5 (110.9–114.1)	112.4 (110.6–114.3)	110.3 (108.5–112.2)	0.03	0.94
≥25	767	120.6 (117.4–124.0)	120.0 (116.9–123.1)	118.6 (115.8–121.6)	119.2 (116.4–122.1)	120.4 (117.5–123.3)	0.89	
Smoking status								
Never smoker	1719	114.3 (112.3–116.3)	113.9 (112.1–115.7)	114.1 (112.4–115.7)	114.3 (112.6–116.1)	113.1 (111.3–114.8)	0.29	0.02
Ever smoker	925	116.1 (112.6–119.7)	115.0 (111.8–118.3)	114.2 (111.0–117.4)	114.4 (111.1–117.7)	112.1 (108.9–115.3)	0.03	
Alcohol consumption								
Non-drinker	920	113.5 (110.3–116.7)	113.2 (110.6–116.0)	115.1 (112.5–117.8)	114.2 (111.6–116.8)	112.6 (110.0–115.3)	0.35	0.06
Current drinker	1732	115.3 (113.3–117.3)	114.8 (112.8–116.8)	113.9 (112.1–115.8)	114.1 (112.2–116.1)	112.9 (111.0–114.9)	0.03	
DBP								
BMI (kg/m^2^)								
<25	1886	71.6 (70.2–73.0)	71.2 (70.0–72.5)	71.3 (70.2–72.5)	71.4 (70.2–72.7)	69.8 (68.5–71.0)	0.01	0.32
≥25	767	76.6 (74.4–78.8)	76.1 (74.0–78.3)	74.8 (72.7–77.1)	76.3 (74.3–78.4)	77.0 (74.9–79.2)	0.61	
Smoking status								
Never smoker	1719	73.2 (71.8–74.6)	72.4 (71.1–73.7)	72.2 (70.9–73.5)	72.6 (71.4–73.9)	72.1 (70.8–73.3)	0.36	0.15
Ever smoker	925	73.0 (70.7–75.4)	73.1 (70.9–75.3)	72.4 (70.3–74.6)	72.9 (70.7–75.3)	70.1 (68.0–72.3)	0.03	
Alcohol consumption								
Non-drinker	920	73.1 (70.9–75.4)	72.6 (70.7–74.5)	73.4 (71.5–75.4)	73.1 (71.4–74.8)	72.4 (70.5–74.3)	0.47	0.12
Current drinker	1732	72.8 (71.4–74.2)	72.9 (71.5–74.2)	71.9 (70.6–73.2)	72.4 (71.1–73.7)	71.3 (70.0–72.6)	0.05	

Abbreviation: CI, confidence interval; SBP, systolic blood pressure; DBP, diastolic blood pressure; BMI, body mass index. ^1^ Adjusted for age, age^2^, BMI (kg/m^2^; continuous), pack years of smoking (never, <10, 10 to <20, ≥20 for men and never, <3, 3 to <6, ≥6 for women), alcohol consumption(none, <15, 15 to <30, ≥30 g/day for men and none, <3.75, 3.75 to <7.5, ≥7.5 g/day for women), marriage status (never, married, divorced or widowed or separated), education level (less than high school graduate, high school graduate, college or above), and menopause status (pre, post) for women. ^2^ We further adjusted for urinary potassium (mmol/L; continuous) in the model. ^3^ We further adjusted for urinary sodium (mmol/L; continuous) in the model.

**Table 4 nutrients-12-03442-t004:** Factor loading values of food groups contributing to urinary sodium-to-creatinine ratio through reduced rank regression analysis.

Food Groups	Factor Loadings ^1^
Positive associations	
Vegetables	+0.461
Soy paste	+0.304
Kimchi	+0.284
Seaweed	+0.233
Inverse associations	
Poultry	−0.295
Pizza	−0.273

^1^ The food groups with the absolute value of factor loading greater than 0.20 were selected.

**Table 5 nutrients-12-03442-t005:** Multivariate-adjusted geometric means (95% CIs) ^1^ of systolic and diastolic blood pressures (mmHg) according to sodium-contributing food score.

	Quintile 1	Quintile 2	Quintile 3	Quintile 4	Quintile 5	*P* Trend
All						
SBP	113.2 (111.6–114.8)	113.1 (111.5–114.7)	113.6 (112.1–115.2)	115.2 (113.5–116.9)	114.5 (112.9–116.2)	0.03
DBP	71.7 (70.6–72.8)	71.8 (70.7–73.0)	72.3 (71.3–73.3)	73.3 (72.2–74.5)	72.9 (71.7–74.1)	0.02
Men						
SBP	119.6 (116.8–122.6)	118.0 (115.1–120.9)	119.4 (116.7–122.1)	120.0 (117.0–123.2)	121.4 (118.5–124.4)	0.07
DBP	74.5 (72.6–76.5)	73.7 (71.9–75.6)	74.5 (72.8–76.2)	75.8 (73.9–77.7)	76.3 (74.3–78.4)	0.02
Women						
SBP	108.4 (106.0–110.9)	109.3 (107.0–111.7)	109.9 (107.6–112.2)	110.8 (108.1–113.5)	109.7 (107.4–112.1)	0.21
DBP	68.9 (67.3–70.5)	69.8 (68.1–71.4)	70.3 (68.7–71.9)	70.8 (69–72.6)	70.1 (68.5–71.8)	0.15

Abbreviation: CI, confidence interval; SBP, systolic blood pressure; DBP, diastolic blood pressure. ^1^ Adjusted for age, age^2^, BMI (kg/m^2^; continuous), pack years of smoking (never, <10, 10 to <20, ≥20 for men and never, <3, 3 to <6, ≥6 for women), alcohol consumption (none, <15, 15 to <30, ≥30 g/day for men and none,<3.75, 3.75 to <7.5, ≥7.5 g/day for women), marriage status (never, married, divorced or widowed or separated), education level (less than high school graduate, high school graduate, college or above), menopause status (pre, post) for women, and urinary potassium (mmol/L; continuous).

**Table 6 nutrients-12-03442-t006:** Multivariate-adjusted geometric means (95% CIs) ^1^ of systolic and diastolic blood pressure according to each food group with a high sodium contributing food score.

	Quartile 1	Quartile 2	Quartile 3	Quartile 4	*P* Trend
SBP					
Vegetables	113.7 (112.2–115.2)	113.6 (112.1–115.1)	113.3 (111.8–114.7)	115.3 (113.8–117.0)	0.02
Soy paste	113.4 (111.8–114.9)	113.2 (111.8–114.7)	114.8 (113.4–116.3)	114.3 (112.8–115.9)	0.23
Kimchi	113.3 (111.8–114.9)	112.7 (111.4–114.1)	114.6 (113.2–116.1)	115.0 (113.4–116.7)	0.007
Seaweed	113.1 (111.7–114.6)	114.2 (112.6–115.7)	113.7 (112.2–115.2)	115.2 (113.6–116.7)	0.02
Poultry	114.0 (112.4–115.5)	114.7 (113.1–116.4)	113.7 (112.2–115.2)	113.9 (112.3–115.4)	0.61
Pizza	114.4 (113.1–115.8)	115.2 (113.1–117.4)	113.1 (111.6–114.8)	113.4 (111.9–115.0)	0.11
DBP					
Vegetables	72.0 (70.9–73.0)	71.9 (70.8–72.9)	72.1 (71.1–73.2)	73.7 (72.5–74.8)	<0.001
Soy paste	71.7 (70.7–72.8)	72.1 (71.1–73.2)	73.3 (72.3–74.4)	72.3 (71.3–73.4)	0.59
Kimchi	72.0 (70.9–73.1)	71.7 (70.7–72.7)	72.8 (71.8–73.9)	73.0 (71.9–74.2)	0.02
Seaweed	71.6 (70.6–72.6)	72.9 (71.8–73.9)	72.3 (71.2–73.4)	73.2 (72.1–74.3)	0.02
Poultry	72.6 (71.5–73.8)	72.8 (71.7–74.0)	72.2 (71.1–73.2)	72.2 (71.1–73.2)	0.25
Pizza	72.5 (71.5–73.4)	74.3 (72.9–75.6)	71.8 (70.7–72.9)	72.3 (71.1–73.5)	0.48

Abbreviation: CI, confidence interval; SBP, systolic blood pressure; DBP, diastolic blood pressure. ^1^ Adjusted for age, age^2^, BMI (kg/m^2^; continuous), pack years of smoking (never, <10, 10 to <20, ≥20 for men and never, <3, 3 to <6, ≥6 for women), alcohol consumption (none, <15, 15 to <30, ≥30 g/day for men and none, <3.75, 3.75 to <7.5, ≥7.5 g/day for women), marriage status (never, married, divorced or widowed or separated), education level (less than high school graduate, high school graduate, college or above), menopause status (pre, post) for women, total energy intake (kcal/d; continuous), and urinary potassium (mmol/L; continuous).

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
