# Peer review of "Urinary Sodium and Potassium Levels and Blood Pressure in Population with High Sodium Intake"

_nutrients, 2020, doi:10.3390/nu12113442_

Round 1
Reviewer 1 Report
The authors have investigated the association between blood pressure and 12-hour urine Na/Cr and K/Cr in a cross-sectional study including 2,653 healthy twin subjects. The authors found that systolic blood pressure was higher is subjects in the highest quintile of Na/Cr, especially in subjects with a BMI < 25. A major limitation of the current study is the use of a 12-hour urine collection, which has not been validated for sodium intake estimation.
Introduction
- Line 67: urin -> urine
Methods
- Blood pressure was measured twice. Which blood pressure did the authors use for analysis
- Was blood pressures measured after a standardized period of rest?
- The authors state that subjects were instructed to void urine around 7 pm and collect urine until the next morning. This method implies that urine collection time can substantially differ between subjects. If the last void before 7 pm was around noon, the total time of urine collection could be almost 20 hours. The authors should provide more details on urine collection time
- Why did the authors use the MDRD for the estimation of eGFR? CKD-EPI is more accurate in the higher regions of kidney function.
- Why did the authors use 12-hour urine collections as this is no validated method for sodium intake estimation?
Results
- What do the authors mean with ‘age-standardized’ characteristics?
- Why did the authors not provide p-values in Table 1?
- Table 3 is very large. The authors should try to reduce the size of the table, for example, by deleting the absolute blood pressure values and 95% CI
- I don’t understand a part of the legend of Table 1, especially the superscript numbers: ‘1Adjusted for age, age2’. Do the authors mean that besides the patient characteristic of interest, the analysis was also corrected for the other patient characteristics that are mentioned in the table legend?
- The authors demonstrate that blood pressure is significantly affected by urine sodium and urine potassium concentrations. Did the authors correct the sodium analysis for urine potassium and vice versa?
- Why did the authors not report food groups that are high in potassium?
- Did the authors also perform the analysis on food vs sodium and food vs blood pressure in subgroups of BMI?
- Can the interacting effect of BMI be explained by higher potassium intake in healthy individuals?
- Did BMI co-localize with other risk factors for high blood pressure?
Discussion
- ‘We observed that urinary sodium-to-creatinine and sodium-to-potassium ratios were positively associated with systolic blood pressure’ This is incorrect as potassium intake was negatively associated with blood pressure
- How do the authors explain that the intake of pizza is associated with lower urine sodium concentrations?
- ‘When age was adjusted for in logistic regression models’. Did the authors perform logistics regression? Do the authors mean linear regression?
- In my opinion, the second paragraph is too long. The authors discuss multiple studies in too much detail. The authors should put more effort into discussing the unique characteristics of the current study such as the association with BMI and the contributing food groups.
- I would suggest deleting the second-last paragraph of the discussion. The authors discuss in detail how sodium intake increases blood pressure, which was not the purpose of the current study.
- Instead, the authors should discuss the use of 12-hour urine collections. How can this method affect the results? This urine collection includes dinner in the evening, could this affect the results as most sodium in consumed then?
Conclusion
The authors have not investigated high blood pressure and should therefore rephrase their conclusion. Their observations are limited to healthy subjects with normal blood pressure
Author Response
Dear Dr. Masanari Kuwabara and reviewer
Thank you for giving us the opportunity to improve and resubmit our manuscript entitled “Urinary sodium and potassium levels and blood pressure in population with high sodium intake”. We responded to comments from editors and reviewers on a point-by-point basis.
We would like to thank the editor and reviewers for the constructive and competent criticism. We believe that the manuscript is much improved in the revised version. We hope that our manuscript will be acceptable for publication in Special issue “Salt in Health and Disease- a Delicate Balance” of Nutrients.
Thank very much for your consideration.
Sincerely,
Jung Eun Lee, Sc.D.
Department of Food and Nutrition,
Seoul National University,
Seoul, Korea
Email: jungelee@snu.ac.kr

Reviewer 2 Report
The authors examined the associations of urinary sodium and potassium levels with cross-sectionally measured blood pressure, using a sample of 2,653 adults participating in the Healthy Twin Study in South Korea. Their results suggest a positive association of sodium and sodium-to-potassium ratio with blood pressure. The authors also captured a dietary pattern contributing to high intake sodium and its subsequent association with blood pressure. This study has some strengths, however, several issues should be clarified and some parts should be added.
- Abstract: It would be more informative to show the quantitative results rather than the p-value.
- Abstract, line 31-32: “The association between urinary sodium-to-creatinine and systolic blood pressure was limited to individuals…” this sentence is not clear. It sounds like the association was only evaluated within participants with BMI<25.
- In the introduction, the authors fail to make a clear link between potassium levels and blood pressure, which has been evaluated in the current study.
- Introduction, line 52, “One possible reason for lack of evidence is..”: This sentence is incomplete. The author should mention what evidence is lacking, perhaps the associations between sodium intake and BP in Koreans?
- Methods, page 3: Hypertension was defined but never used in the analysis. So did the authors also try the analysis evaluating associations between sodium, potassium, and prevalence hypertension?
- Methods: Did the information on the use of anti-hypertensive medications collected in this study? If so, this variable should be considered in the analysis. If not, it should be acknowledged as a limitation of the study.
- Methods, page 4: The authors should clarify how the effect modification was examined, by adding the interaction terms between exposure and modifiers, or conducting stratified analyses?
- Methods: More information about what the RRR algorithm does can be useful for some readers.
- Results, Table 1: Please add the range of sodium-to-creatinine ratio in each quintile in Table 1.
- Results: Information on potassium-to-creatinine ratio, and sodium-to-potassium ratio should also be included in Table 1.
- Results: Again, the ranges of exposures in different quintiles should be added to table 2.
- Results: The authors should clarify whether the same quintiles were used for men and for women.
- Did the authors adjusted for potassium in the model evaluating the association between sodium and BP? Similarly, was sodium adjusted in the model assessing the association between potassium and BP?
- Results, page 10, “A higher sodium-contributing food score was associated 211 with increasing blood pressure in men (p for trend <0.06): The authors mentioned that the significant level of 0.05 was used. This way the association should not be considered statistically significant. Additionally, the authors should state the positive associations between food score and SBP and DBP in the entire population.
- Discussion, first sentence: I am wondering why the authors only highlight the associations found in men given the significant associations were also observed in women.
- Discussion: The authors should clearly highlight the new information that can be added to the existing reference by this study given tons of previous studies have reported the associations between sodium and BP, particularly in Koreans.
- The discussion about the association between potassium and BP and its underlying biological mechanism is also missing in the discussion.
Author Response

(The authors gave the same response as above.)

Reviewer 3 Report
In the present study Song and coauthors evaluated the association of sodium/creatinine ratio and potassium/creatinine ration in Korean adults who had participated in the Healthy twin study. They found that higher urinary sodium/creatinine ratio is associated with higer blood pressure values. Although the study is well designed and also well written I believe that the study does not add much to current knowledge. Furthermore urinary sodium/creatinine and potassium/creatinine ratio were measured on a single urinary sample in patients who were not on a fixed-dose sodium chloride diet. Finally although hypertensive patients were excluded there is no mention to patients with diuretics or CKD
Author Response
In the present study Song and coauthors evaluated the association of sodium/creatinine ratio and potassium/creatinine ration in Korean adults who had participated in the Healthy twin study. They found that higher urinary sodium/creatinine ratio is associated with higer blood pressure values. Although the study is well designed and also well written I believe that the study does not add much to current knowledge. Furthermore urinary sodium/creatinine and potassium/creatinine ratio were measured on a single urinary sample in patients who were not on a fixed-dose sodium chloride diet. Finally although hypertensive patients were excluded there is no mention to patients with diuretics or CKD
We appreciated the reviewer’s point. The large sample size of this study may contribute to understanding of how sodium and potassium play roles in blood pressure increase. Also only a few studies explored urinary sodium and potassium among populations with high sodium intake. We believe that this study will be a good basis for health-policy making and need of moderation in sodium consumption in Asian population.

Round 2
Reviewer 1 Report
The authors have added data of a previous validation study. However, they have validated the 12-hour urine against 24-hour instead of the actual intake. The authors should therefore add a limitation to the discussion that the 12-hour urine samples have been limitedly validated and have not been compared to measured intake.
The readability of the manuscript is still poor due to very large tables. I would suggest to substantially reduce the size of the tables. For example, why provide data on male and female subjects in all tables, while this is not a major research question and only Table 5 demonstrates substantial differences between males and females. I would suggest showing data for the entire study population and possibly provide gender data in the supplemental information.
The authors explained their standardization for age. What is the added value of this method? If I am correct, age-standardization is mostly used to compare population characteristics between two populations that significantly differ in age. However, the current study investigated one population and I, therefore, do not see the added value. I would therefore suggest to present data that are not standardized by age and report that age differs between intake quintiles, and correct the regression analyses for age (just like the authors do already).
- ‘We observed that urinary sodium-to-creatinine and sodium-to-potassium ratios were positively associated with systolic blood pressure’ This is incorrect as potassium intake was negatively associated with blood pressure.
We observed that both sodium-to-creatinine and sodium-to-potassium ratios were positively associated with systolic blood pressure (Table 2).
I don’t agree with this statement. With higher sodium intake, blood pressure increases (positive association). With higher potassium intake, blood pressure decrease (negative association)
Author Response
Comments from Reviewer #1:
The authors have added data of a previous validation study. However, they have validated the 12-hour urine against 24-hour instead of the actual intake. The authors should therefore add a limitation to the discussion that the 12-hour urine samples have been limitedly validated and have not been compared to measured intake.
We have added the limitation about the validation in the discussion as follows:
“We did not take into account dietary intake of sodium or correlation between urinary sdoiumintake and dietary sodium intake.”
The readability of the manuscript is still poor due to very large tables. I would suggest to substantially reduce the size of the tables. For example, why provide data on male and female subjects in all tables, while this is not a major research question and only Table 5 demonstrates substantial differences between males and females. I would suggest showing data for the entire study population and possibly provide gender data in the supplemental information.
We have shorten the table1-2 by moving the results presented by sex to the supplement materials.
“The baseline characteristics by sex was presented in the Supplementary Table 1.”
The authors explained their standardization for age. What is the added value of this method? If I am correct, age-standardization is mostly used to compare population characteristics between two populations that significantly differ in age. However, the current study investigated one population and I, therefore, do not see the added value. I would therefore suggest to present data that are not standardized by age and report that age differs between intake quintiles, and correct the regression analyses for age (just like the authors do already).
We revised the Table1 as presented the crude characteristics according to quintiles of sodium-to-creatinine ratio, potassium-to-creatinine ratio, or sodium-to-potassium ratio among men and women combined. Age was adjusted in the generalized linear model.
- ‘We observed that urinary sodium-to-creatinine and sodium-to-potassium ratios were positively associated with systolic blood pressure’. This is incorrect as potassium intake was negatively associated with blood pressure.
We observed that both sodium-to-creatinine and sodium-to-potassium ratios were positively associated with systolic blood pressure (Table 2).
I don’t agree with this statement. With higher sodium intake, blood pressure increases (positive association). With higher potassium intake, blood pressure decrease (negative association)
We appreciated the reviewer’s point. We have revised the sentence as follows:
“We observed that urinary sodium-to-creatinine ratio was positively associated with systolic blood pressure, while potassium-to-creatinine was negatively associated with systolic blood pressure in men and women combined”

Reviewer 2 Report
The authors addressed all my comments.
Author Response
We appreciated the reviewer’s comments.